



**Impact of aerosol size distribution on extinction and spectral dependence of radiances measured by the OMPS Limb profiler instrument**

Zhong Chen[1], Pawan K. Bhartia[2], Robert Loughman[3], Peter Colarco[2]

[1]Science Systems and Applications, Inc., Lanham, Maryland, 20706, USA

[2]NASA Goddard Space Flight Center, Greenbelt, Maryland 20771, USA

[3]Department of Atmospheric and Planetary Sciences, Hampton University, Hampton, Virginia, USA

Correspondence to: Zhong Chen (zhong.chen@ssaihq.com)

**Abstract**

The Ozone Mapping and Profiler Suite Limb Profiler (OMPS/LP) has been flying on the Suomi NPP satellite since Oct 2011. It is designed to produce ozone and aerosol vertical profiles at ~2 km vertical resolution over the entire sunlit globe. The current operational (V1) aerosol extinction retrieval algorithm assumes a bimodal lognormal aerosol size

distribution (ASD) whose parameters were derived from in situ data taken from an aircraft. In this paper we discuss the impact on the retrieval of using an ASD derived by the Community Aerosol and Radiation Model for Atmospheres (CARMA). We find that the impact of ASD on the retrieved extinctions varies strongly with the underlying reflectivity of the scene, and the functional form of this variation is very different at

different scattering angles. We also evaluate how well the two ASDs perform in explaining the spectral dependence of Aerosol Scattering Index (ASI); a dimensionless quantity that we derive from the measured radiances by subtracting out the Rayleigh contribution. ASI is easier to interpret than radiances themselves and serves as our measurement vector. The results show that even though the two ASDs produce very

different aerosol scattering phase function values at small and large scattering angles, the effect of the ASD on the spectral dependence of ASI is significant only at small angles. This implies that while OMPS/LP measurements have some information to evaluate the ASDs, they are most effective only at small scattering angles, which for LP measurement geometry occur only in the northern hemisphere. Our analysis suggests that overall

CARMA ASD does a better job in explaining the spectral dependence of measured ASI than the ASD used in the operational V1 algorithm.



## 1. Introduction

Accurate estimation of stratospheric aerosol is important because aerosols in the stratosphere have an important influence on climate variability and also play an important role in the chemical and dynamic processes related to ozone destruction in the

stratosphere. Therefore, long-term measurement of the distribution of aerosols is necessary for a better understanding of stratospheric processes.

The Ozone Mapping and Profiler Suite Limb Profiler (OMPS/LP) is one of three OMPS instruments onboard the Suomi National Polar-orbiting Partnership (S-NPP) satellite (Flynn et al., 2007). S-NPP was launched in October 2011, into a sun-synchronous polar

orbit. The local time of the ascending node of the S-NPP orbit is 13:30. The LP instrument collects limb scattered radiance data and solar irradiance data on a 2-D charge coupled device (CCD) array over a wide spectral range (290-1000 nm) and a wide vertical range (0-80 km) through three parallel vertical slits. These spectra are primarily used to retrieve vertical profiles of ozone (Rault and Loughman, 2013), aerosol extinction

coefficient (Loughman at al., 2017), and also cloud-top height (Chen et al., 2016). Jaross et al. (2013) provides more details about the OMPS/LP instrument design and capabilities.

Instruments that measure scattered radiation need to assume some form of aerosol size distribution (ASD) to convert their measured information into aerosol extinction. These

instruments include limb scattered instruments such as SCIAMACHY (von Savigny et al., 2015), OSIRIS (Bourassa et al., 2008), OMPS/LP (Loughman at al., 2017), and space and ground-based lidars, e.g., CALIOP (Winker et al., 2009). Aerosol Chemical transport models, such as the Goddard Chemistry, Aerosol, Radiation, and Transport (GOCART) module (Colarco et al., 2010) and the GEOS-Chem (Bey et al., 2001), typically provide

aerosol mass density as a function of altitude. In order to convert these data into extinction, they also need to assume an ASD. By contrast, instruments that employ solar, lunar, and stellar occultation techniques such as SAGE II (Chu et al., 1989), SAGE III (Thomason et al., 2010) and GOMOS (Bertaux et al., 2010) can estimate extinction directly from their transmission measurements without assuming an ASD.

The purpose of this paper is to examine the sensitivity of the OMPS/LP V1 aerosol algorithm (Loughman et al., 2017) to ASD. In this study we examine how the results





change if we replace the ASD assumed in V1 with an ASD derived from the CARMA model (Colarco et al., 2003, 2014). We then examine whether the spectral dependence of LP radiances have information that can help us select between these two ASDs. We expect the results of such studies to help select appropriate ASDs for future version of the

retrieval algorithm.

## 2. Aerosol Microphysical Models

The current OMPS/LP retrieves stratospheric aerosol extinction profiles by assuming a bimodal lognormal (BMLN) size distribution. The fine and coarse mode size parameters

of this distribution (see Table 1) are based on ER-2 measurements in August, 1991, at 36°N and 121°W and at 16.5km (Pueschel et al., 1994), with the coarse mode fraction adjusted to provide an Angstrom Exponent (defined in Eq, (1)) of 2.0. This is the mean value of AE at altitude 20 km estimated from SAGE II (version 7.0) aerosol extinction data (Damadeo et al., 2013) at 525 and 1020nm taken during the period 2000-2005, when

the stratosphere was relatively clean and roughly similar to the present day stratosphere as shown in Figure 1.

**Table 1.** Bimodal size distribution used in OMPS/LP version 1 aerosol extinction retrieval

| AE | $r_{eff}$ (µm) | $f_c$ | $r_i$ (µm) | $\sigma_i$ |
|---|---|---|---|---|
| 2.0 | 0.14 | 0.003 | 0.09, 0.32 | 1.4,1.6 |

$$AE = -\frac{\ln[E(\lambda_1)] - \ln[E(\lambda_2)]}{\ln(\lambda_1) - \ln(\lambda_2)} \qquad (1)$$

where, $E$ is the aerosol extinction at wavelength $\lambda$.

To test the sensitivity of the OMPS/LP aerosol algorithm to ASD, we examine the data produced by the Community Aerosol and Radiation Model for Atmospheres (CARMA) model. CARMA is a general-purpose sectional microphysics model that has been used to

study a wide variety of aerosols in planetary atmospheres (Toon et al., 1979, 1988; Turco et al., 1979; Bardeen et al., 2008; Colarco et al., 2003, 2014; English et al., 2011, 2012; Yu et al., 2015).



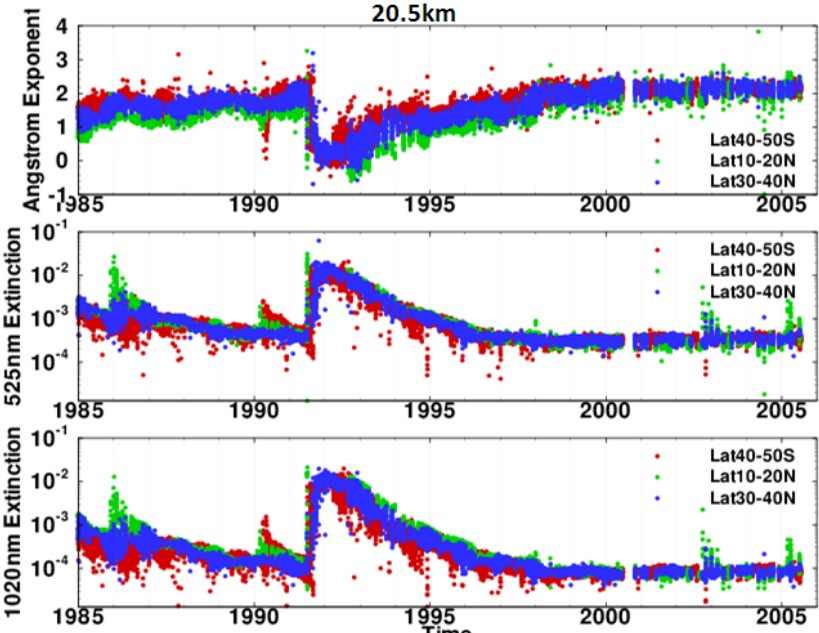

**Figure 1.** Time series of Angstrom Exponent (AE) (top) derived from the aerosol extinction coefficient at 525 nm (middle) and 1020 nm (bottom) at 20 km altitude. This figure shows SAGE version 7 data for the 40S–50S (red), 10-20N (green) and 30-40N (blue) latitude bins during the period 1985 - 2005. While the Pinatubo eruption in 1991 produced a significant decrease in AE, the smaller Anatahan volcano eruptions in 2003 and 2005 (visible in the extinction time series) did not affect AE values.

Figure 2a shows the sample of particle number size distribution simulated by the CARMA model run in the GEOS-5 system (Colarco et al. 2014). The model was run at a global ~1 degree horizontal resolution with 72 vertical levels from the surface to the ~85 km model top and included precursor emissions for anthropogenic sulfates, degassing volcanoes (but not explosive eruptions), and the naturally occurring background stratospheric aerosol layer. The sulfate aerosol mechanism used here is as in English et al. (2011, 2012). The particle size distribution is simulated using 22 particle size bins which cover dry radii from 0.000267 micron to 2.79 microns, and particle growth occurs by homogeneous nucleation of sulfuric acid vapor, condensation of sulfuric acid onto aerosols, and coagulation. The model was a climate model run (i.e., free-running atmosphere) driven by observed sea surface temperatures for the period 1990 - 1993. Results in Figure 2a show the summertime (June-July-August) climatological mean particle size distribution at 20 km altitude at Laramie, Wyoming (41 N, 105 W),





which is chosen as a validation point because of the long-term availability of balloon observations at that location (Deshler et al. 2003).

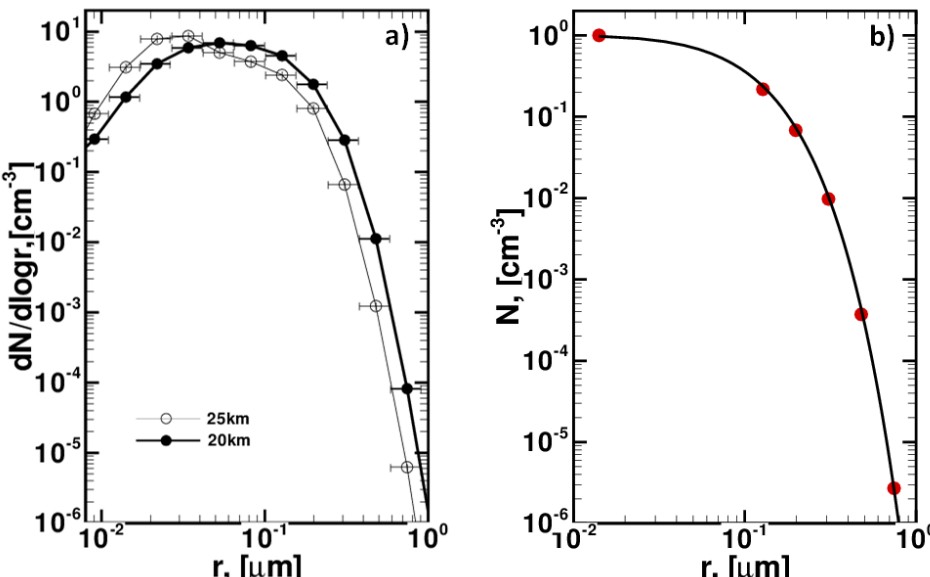

**Figure 2.** Stratospheric aerosol size distribution (ASD) as a function of radius. (a) ASDs at 20 km and 25 km from the CARMA microphysical model shown as number density in 11 size bins in the aerosol radius range between 0.009 and 1 μm. The widths of the CARMA size bins are also shown as horizontal lines. The curves do not follow the lognormal distribution often used to model ASDs. (b) A Gamma distribution fit to the cumulative number density (N>r) calculated from the data shown in (a) and normalized to 1 at the smallest radius. The black line represents the fit to data using the fitted parameters. The CARMA data are shown as red dots.

The aerosol optical properties can be calculated based on Mie theory for each bin of the size distribution, but in most applications an analytic model is instead fit to the discrete bin populations. Visual examination of CARMA data (see Figure 2a) shows that it follows a power law rather than a lognormal distribution at the shorter radii. So, we have selected the widely used Gamma distribution (e.g., Chylek et al., 1992) to fit a mathematical function to CARMA data. This function is described in Eq. (2).

$$n(r) = \frac{\beta^\alpha r^{\alpha-1}}{\Gamma(\alpha)} \exp(-r\beta) \qquad (2)$$

where α and β are the fitting parameters, and Γ is Euler's Gamma function, defined as:

$$\Gamma(\alpha) = \int_0^\infty t^{\alpha-1} \exp(-t)dt \,, \qquad (3)$$





At small radii this function follows a power law and at large radii an exponential function. In contrast to the BMLN, which has 5 adjustable parameters, this function has only two such parameters, the shape parameter α and the scale parameter β with a unique relationship to the effective radius (Chylek et al., 1992):

$$r_{eff} = \frac{\int_0^\infty r^3 n(r)dr}{\int_0^\infty r^2 n(r)dr} = \frac{(\alpha + 2)}{\beta}$$ (4)

To fit the Gamma distribution (GD) to CARMA data, which is provided in coarse radii bins, we calculate cumulative aerosol size distribution,

$$N(>r) = \int_r^{r_{max}} n(r)dr$$ (5)

where, $N(>r)$ represents the concentration of all particles larger than r. The integral is

performed over a range of sizes from $r_{min}$ to $r_{max}$. The two parameters of the GD are determined by fitting the cumulative distribution function (CDF) of Eq. (5) by a Levenberg-Marquardt nonlinear least squares regression algorithm. The scattering cross sections and phase functions are then calculated using Mie theory assuming spherical particles of refractive index of 1.448 + 0i, which is same as that assumed in V1 OMPS/

LP aerosol algorithm. Hereafter, the resultant ASD from the CARMA data will be labeled as CARMA ASD, while the ASD assumed in V1 based on aircraft measurements by Pueschel et al. (1994) will be labeled as Pueschel ASD.

Figure 2b shows that the GD fits the CARMA data quite well. All CARMA points are

almost exactly reproduced by the Gamma function suggesting that CARAM data are the GD distribution. Figure 3 compares CARMA and Pueschel ASDs, which are plotted as dN/dlogr vs r in log-log scale (here log is the logarithm to base 10). The best-fit parameters, together with the calculated values of AE and $r_{eff}$ are given in Table 2.

**Table 2.** Gamma-CARMA size distribution used in this work

| AE | $r_{eff}$ (µm) | $\alpha$ | $\beta$ |
|---|---|---|---|
| 2.1 | 0.18 | 1.8 | 20.5 |





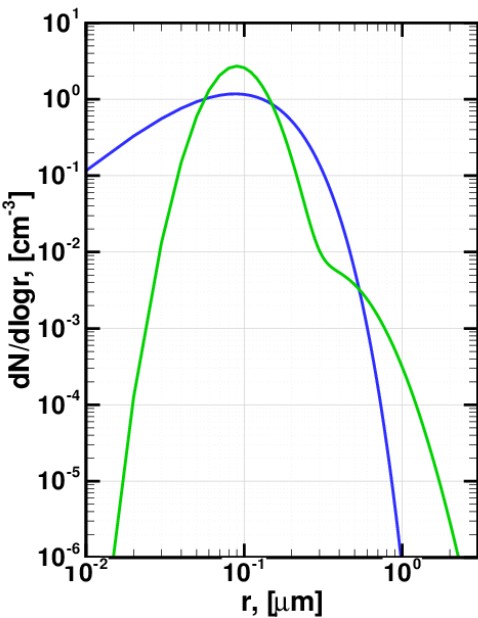

**Figure 3.** Comparison of number densities from the CARMA Gamma distribution fit (blue) with Pueschel (green). The latter distribution has more fine mode particles near 0.1 μm as well as more coarse mode particles with radii >0.5 μm.

We find that the key difference between the two ASDs is that the Pueschel distribution has larger dN/dlogr values at 0.1 micron, which causes the derived aerosol scattering phase function $P(\Theta)$, shown in Figure 4, to be more "Rayleigh-like" at large single scattering angle $\Theta$, i.e., closer to the Rayleigh $P(\Theta)$. This result occurs despite the fact that the Pueschel ASD has a secondary coarse mode peak that should make the $P(\Theta)$ less Rayleigh-like, indicating that at 675 nm $P(\Theta)$ is very sensitive to the dN/dlogr values near 0.1 micron. The differences between the two $P(\Theta)$s vary considerably in both sign and magnitude as a function of $\Theta$. The largest fractional differences (CARMA values 40% less than Pueschel) occur at $\Theta$ >120º, which can lead to a factor of 2.5 larger extinction values at low effective reflectivities $\rho$ (see Section 3), where the derived extinctions are roughly inversely proportional to the $P(\Theta)$, as discussed by Loughman et al. (2017).

OMPS/LP measurements cover a wide range of scattering angles. Figure 5 shows the variation of $\Theta$ with latitude for solstice conditions. Note that high values of $\Theta$ are always



observed in the Southern Hemisphere, while low values of Θ are observed in the Northern Hemisphere. The impact of this sampling is discussed in Section 3.

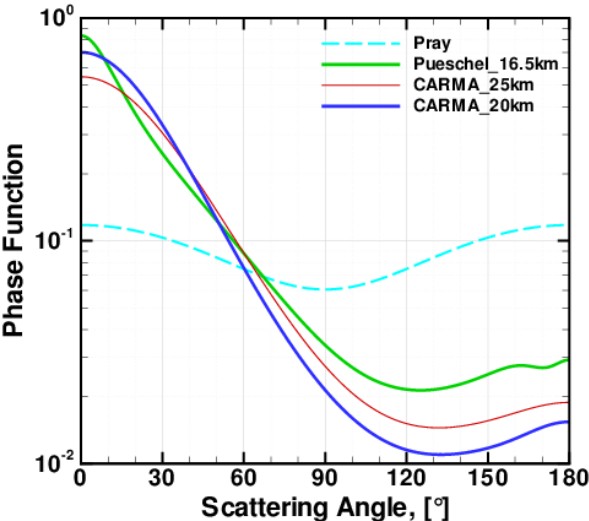

**Figure 4.** Comparison of 675nm P(Θ) between Pueschel at 16.5 km (green) and CARMA at 20 km (blue) and 25 km (red). The Rayleigh P(Θ) is shown as a dashed line. The Pueschel P(Θ) is more Rayleigh-like despite having more coarse particles (>0.5 μm) than CARMA. This is because particles with radii near 0.1 μm have larger influence on P(Θ).

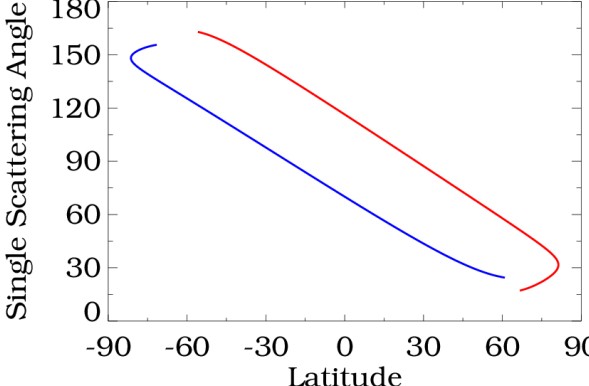

**Figure 5.** Variation of scattering angle vs. latitude for OMPS/LP measurements on June 22 (red) and December 22 (blue).

Figure 6 shows the sensitivity of the AE and $r_{eff}$ to changes in the two fitting parameters.

The new GD model produces an AE of 2.0 and a $r_{eff}$ of 0.18 $\mu$m. These values match the




average values determined from SAGE II version 7.0 data (Thomason et al., 2008; Damadeo et al., 2013) during the 2000-2005 period.

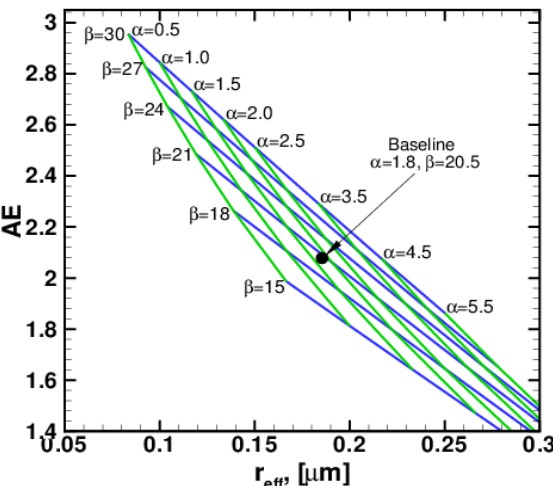

**Figure 6.** Angstrom Exponent (AE) as a function of effective radius $r_{eff}$ calculated from the Gamma size distribution model for different model parameters. Note that AE by itself does not provide information to determine both α and β, and hence $r_{eff}$.

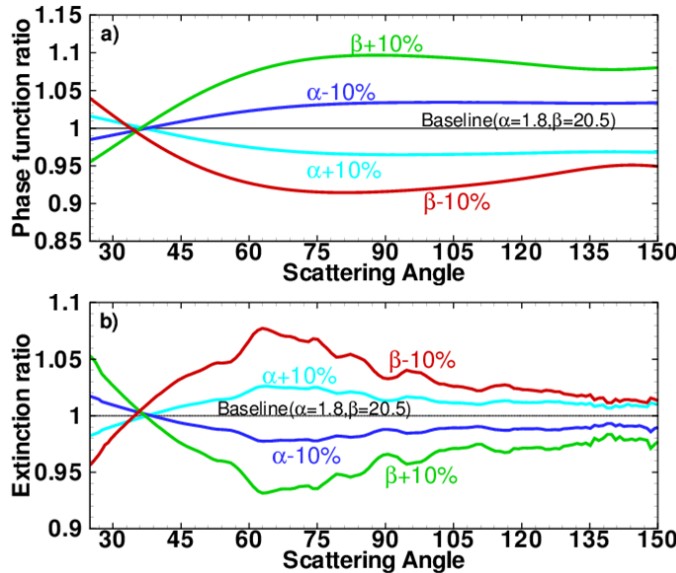

**Figure 7.** Phase function P(Θ) ratios (a) and retrieved extinction ratios (b) as a function of single scattering angle Θ for ±10% change in Gamma distribution parameters. Note that the two curves are roughly anti-correlated, but the fractional change in extinction is about half of the change in P(Θ).




Figure 7a shows the impact on P(Θ) of changing the mode parameters by ±10% relative to the baseline mode. As a consequence, the changes in P(Θ) (Figure 7a) lead to significant changes in aerosol extinctions as shown in Figure 7b.

## 3. Sensitivity of Retrieved Extinctions to ASD

To evaluate the impact of the ASD changes on aerosol retrievals, we incorporated the two ASDs derived from CARMA and Pueschel into the LP V1.0 algorithm and reprocessed OMPS/LP data for one month before and after the Calbuco volcano eruption. The eruption of Calbuco occurred in Chile (Latitude = 41.3°S, Longitude = 72.6°W) on April 22, 2015, and had an impact on the global stratospheric aerosol distribution.

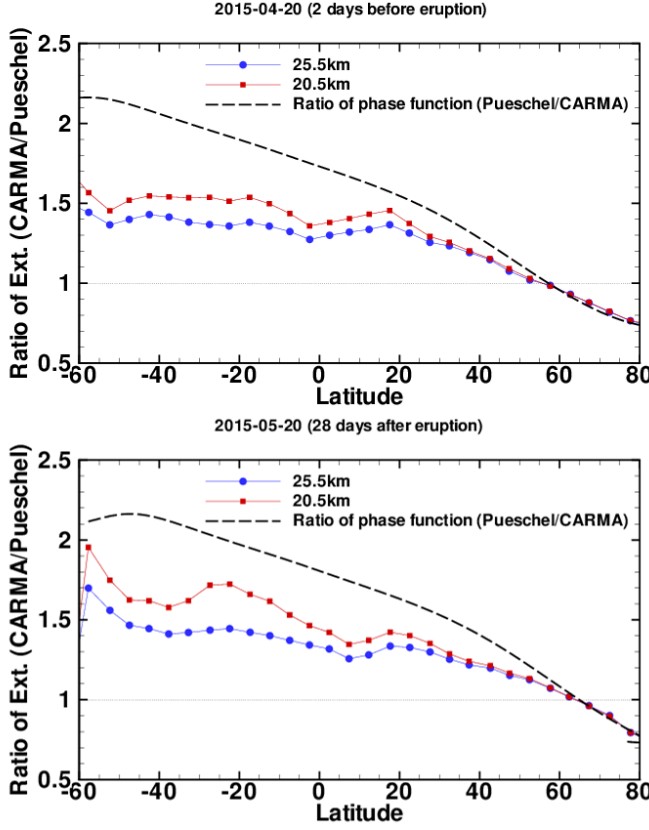

**Figure 8.** Ratio of aerosol extinction zonal means retrieved using two size models as a function of latitude at 20.5 km and 25.5 km. Both plots are for one day of data taken 2 days before (top) and 28 days after (bottom) the Calbuco volcanic eruption. Dashed line shows the inverse of phase function (P(Θ)) ratio. Note that the extinction ratios are smaller than P(Θ) ratios and vary with altitude, but they are not significantly affected by the presence of volcanic aerosols.




Figure 8 compares the zonal mean ratios of aerosol extinctions derived from the two ASDs at 20.5 km and 25.5 km in 5° zonal latitude bands for both volcanic and non-volcanic cases. For comparison, the inverse of P(Θ) ratio is also plotted in Figure 8 as a function of latitude. It can be seen that the mean ratios of aerosol extinction, which vary with altitude, are smaller than the ratio of P(Θ)s for most latitudes. The divergence in extinctions derived from the two ASDs is not as large as the divergence from two P(Θ)s, which is influenced by the scattering angle variation shown in Figure 5. This is duo to multiple scattering which smears the effect of the phase function. There is very little difference in extinction ratios between volcanic and non-volcanic cases.

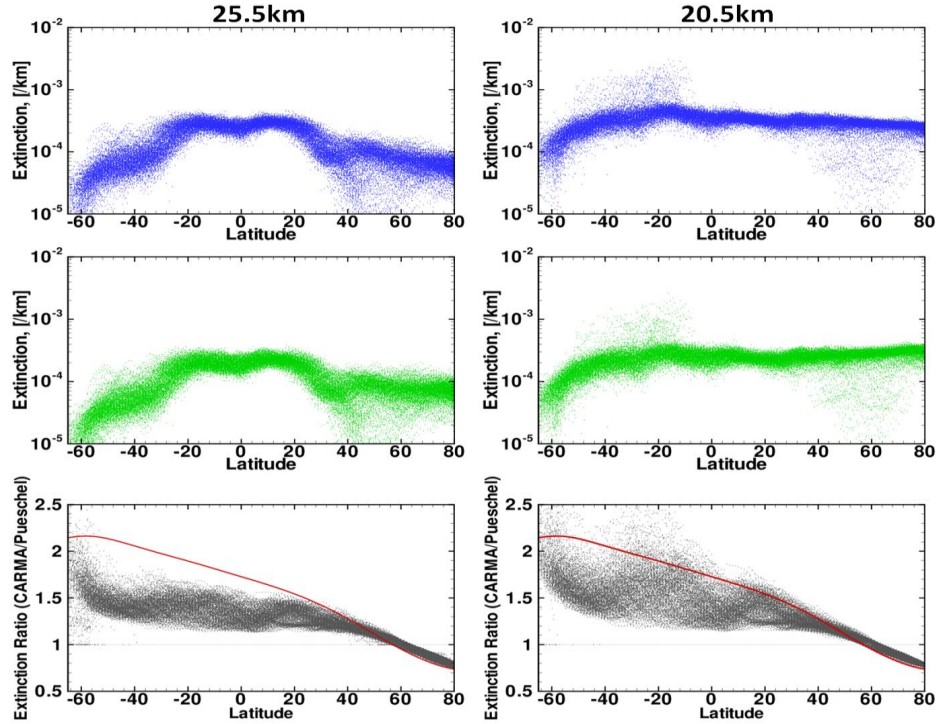

**Figure 9.** Scatter diagrams of retrieved aerosol extinctions for the CARMA model (blue) and the Pueschel model (green) at 25.5 km (left panel) and 20.5km (right panel) for entire month during the Calbuco period April 21 ~May 20, 2015. The black dots in the bottom panel show extinction ratios (CARMA/Pueschel), and the red lines shows the inverse of P(Θ) ratio (Pueschel/CARMA). The ratio of extinctions has large variability at a given latitude, though the P(Θ) ratios do not.

Figure 9 shows scatter diagrams of retrieved aerosol extinctions for the CARMA ASD (blue) and from Pueschel ASD (green) as well as their ratios η (black) at 20.5 and 25.5 km as a function of latitude for the entire month of data during the Calbuco period April





21 ~May 20, 2015. Although the extinctions from the two ASDs exhibit the same pattern of aerosol extinction, η generally decreases with increasing latitude. The values of η are larger than 1 from southern to northern latitudes (< 60ºN). At high northern latitudes (> 60ºN), the values of η at 20.5 km are 75% lower than high southern latitudes. The fact

that the values of η are larger than 1 can be explained by noting that the CARMA ASD has smaller values of P(Θ) in backward scattering directions (see Figure 4), which yields larger values of aerosol extinction at southern low to mid-latitudes for OMPS/LP measurements (see Figure 5).

A notable feature of Figure 9 is the large variation of the extinction ratio η in the

Southern Hemisphere (SH), which is correlated mainly with the variation of effective reflectivity ρ. The ρ value is sometimes called the "Lambert-equivalent reflectivity". It does not equal the true reflectivity of the surface, since the scene generally contains clouds, aerosols, etc. As discussed in Loughman et al. (2017), the effect of Rayleigh scattering and aerosol scattering on radiances is not strictly additive. Rayleigh scattering

also attenuates aerosol scattering, which reduces the measured radiance. This effect increases at lower altitudes, ultimately making the radiances insensitive to aerosol scattering. This behavior is further illustrated in Figure 10. In Figure 10, extinction ratios and values of ρ are binned in 13 latitude bands throughout the same time period shown in Figure 9, so that the influence of ρ on η is clearly discernable.

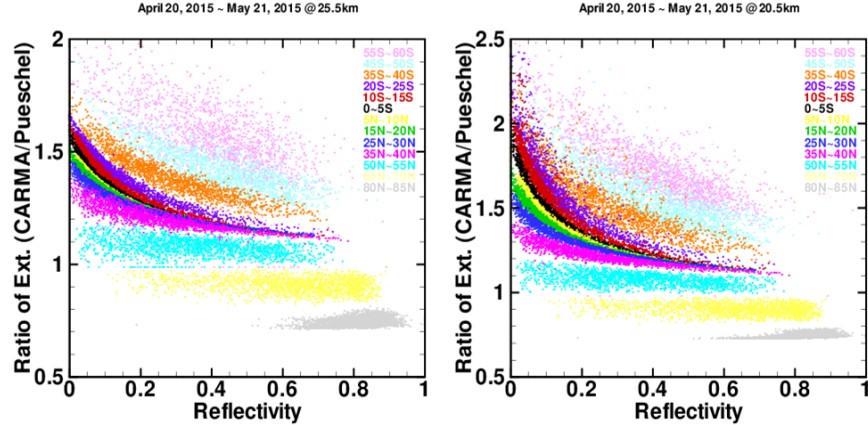


**Figure 10.** Scatter plots of extinction ratio η (CARMA/Pueschel) as a function of effective reflectivity ρ for different latitude bins at 20.5 km (right) and 25.5 km (left). The figure shows that the extinction ratios vary non-linearly with effective reflectivity and the shape of the function changes considerably with latitude and altitude.

## 4. Sensitivity of Spectral Dependence of Radiances to ASD

In the OMPS/LP aerosol algorithm we use a dimensionless quantity, which we call the Aerosol Scattering Index (ASI), as the measurement vector for the retrieval. The measured ASI at wavelength λ and altitude z, is defined as follows:

$$ASI_m(\lambda, z) = [I_m(\lambda, z) - I_0(\lambda, z)] / I_0(\lambda, z) \qquad (6)$$

where $I_m$ is the measured radiance normalized at 40.5 km, and $I_0$ is the calculated radiance for a pure Rayleigh atmosphere, similarly normalized. Though the normalization reduces the effect of diffuse upwelling radiation (DUR) considerably, there are second order effects present that make ASI sensitive to ρ at altitudes where there are aerosols.

10 This occurs because DUR is scattered by the aerosols at an average Θ close to 90˚, while the direct solar radiation is scattered at a range of Θs from 20˚ ~ 140˚ varying with latitude (see Figure 5). As noted previously, Rayleigh scattering attenuation of aerosol scattering increases at lower altitudes, so that ASI also becomes insensitive to aerosol scattering.

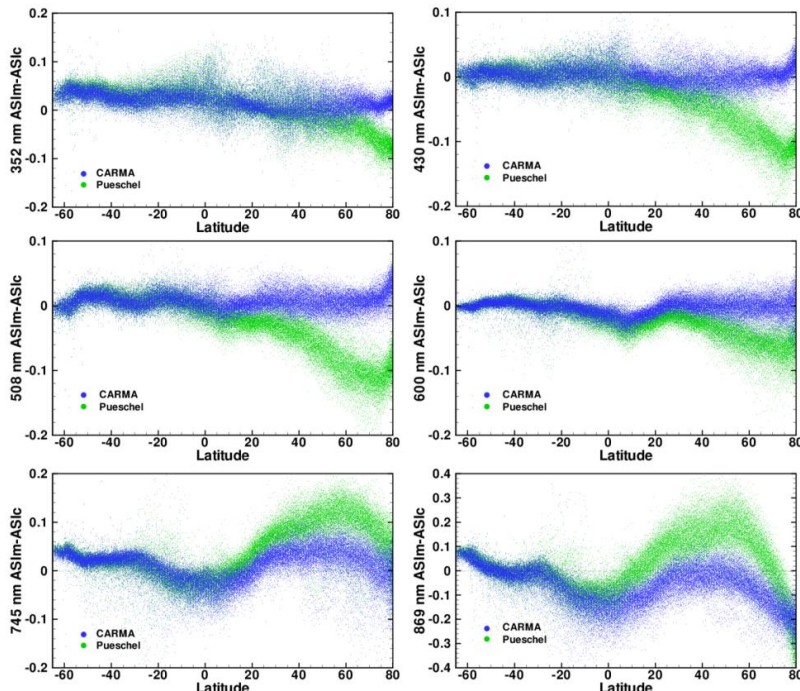

**Figure 11.** ASI residuals (the measured ASI - the calculated ASI) at 20.5 km as a function of latitude for wavelengths at 352nm, 340nm, 508nm and 600nm. CARMA ASD (blue dots) does a better job in explaining the measured ASI relationship than the Pueschel ASD (green dots).





For singly scattered (SS) radiances, assuming that the attenuation of SS radiance along the LOS is small, ASI is proportional to the product of aerosol extinction $E$, and P($\Theta$). So, in this approximation the spectral dependence of ASI should be determined by the spectral dependence of $E$*P($\Theta$), which is determined by ASD. Hence, if the assumed

ASD is correct, the measured and calculated spectral dependence of ASI should be the same, and vice-versa.

In Figure 11, ASI residuals (the measured ASI - the calculated ASI) at 20.5 km as a function of latitude for wavelengths at 352nm, 340nm, 508nm and 600nm are plotted for the entire month during the Calbuco period April 21 ~ May 20, 2015. Figure 12 shows

the ratio of ASI(745nm)/ASI(508nm) at 20.5 km as a function of latitude. From Figures 11 and 12, it is evident that the calculated ASIs from the CARMA ASD agree better with the measurements than the ASI from the Pueschel ASD. Such studies with other candidate ASDs can help select between various ASDs. However, as Figures 11 and 12 show, this technique works best in the Northern Hemisphere where the scattering angles

are small for OMPS/LP measurements. In the Southern Hemisphere where the scattering angles are large, the relative uncertainties in P($\Theta$) tend to be larger.

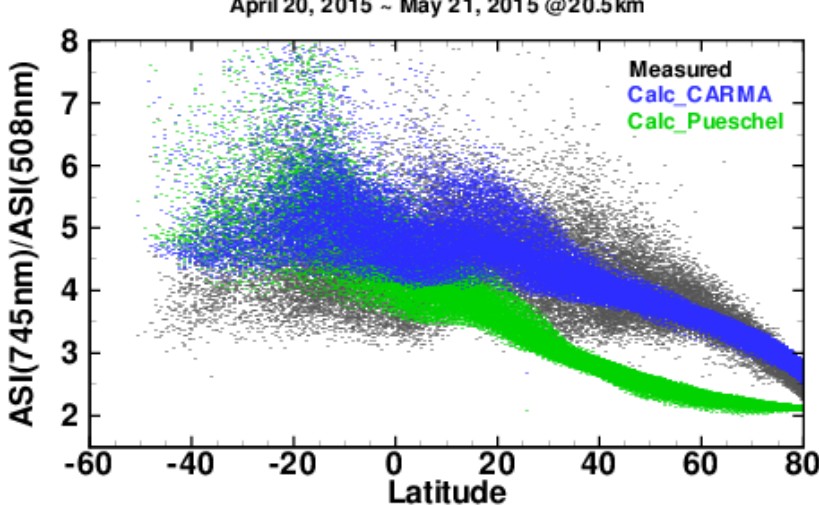

**Figure 12.** Ratio of ASI(745nm)/ASI(508nm) at 20.5 km as a function of latitude. Calculated
values using the CARMA model ASD (blue dots) are more effective in explaining the measured ASI ratios (black dots) than the Pueschel ASD (green dots).





## 5. Summary and Conclusions

Our results show that P(Θ) is very sensitive to the assumed aerosol particle number density near a particle radius of 0.1 micron. Since these values are poorly characterized by in situ measurements, we have used ASD derived from a microphysical model

CARMA to process a small subset of OMPS/LP data to assess the sensitivity of our aerosol retrieval algorithm to ASD. We find that P(Θ) derived from CARMA disagrees substantially with our assumed value at very small and large Θs, but agrees well near Θ = 60˚. The relative difference between the two is largest at Θ = 120˚, where it reaches a factor of 2.5. However, unlike nadir-viewing scattering instruments where 1% error in the

value of P(Θ) at a given Θ produces -1% error in retrieved extinction, for limb scattering the error in extinction is considerably smaller and decreases in magnitude with increase in reflectance of the underlying scene. The functional form of this dependence varies considerably with Θ. Finally, we find that the spectral dependence of limb radiances is not significantly affected by ASD, except at small Θ. Based on this analysis we find that

CARMA ASD does a better job in explaining the spectral dependence of measured radiances. This technique can be used to evaluate other candidate ASDs. We plan to use the data from the recently operational ISS/SAGE III, which uses the solar occultation technique to derive aerosol extinction without assuming ASD, to validate the conclusions derived in this paper.


## Acknowledgements

We thank the OMPS/LP team at NASA Goddard and Science Systems and applications, Inc. (SSAI) for help in producing the data used in this study. We also would like to thank Matthew DeLand for his valuable comments and Tong Zhu for her technical supports.

Zhong Chen was supported by NASA contract NNG17HP01C.

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
