# Peer review of "Impact of aerosol size distribution on extinction and spectral dependence of"

_Atmospheric Measurement Techniques, 2018_

## Referee Comment (RC1) · Anonymous Referee #1 · 13 Mar 2018

**1   General Comments**

This article discusses the retrieval of stratospheric aerosol extinction profiles using the OMPS-LP measurements. The authors use a gamma particle size distribution derived from the CARMA model instead of the standard lognormal assumption, and it is found that this helps to improve the spectral response of the modelled signal at 20.5 km. The approach is a novel one and valuable to the limb scattering aerosol retrieval community. The writing is concise, and the material is generally well explained; after the following minor edits I recommend publication.

[Figure]

**2 Specific Comments**

Some additional information on how and why the CARMA ASD was chosen would be beneficial. It is not clear what sampling is used to derive the parameters in Table 2. What years are the June-July-August data from, what altitudes are used, etc. Is a single GD chosen due to retrieval requirements, or other reasons? Why is sampling near Laramie important if the balloon data is not compared against?

**Figure 1**: Usually the majority of the increase in extinction is attributed to Ruang/Reventador in late 2002 (from the figure it appears the increase starts before 2003) and Manam in 2005 (eg. Vernier et al, 2011). Is there a reason the increase is attributed to Anatahan here?

**Figure 2**: Why is only the 20 km altitude shown in panel B? Also, why are only select CARMA radii used as comparison points (red dots) in panel B and not all of them?

**Page 5, Line 15-16**: It is not clear that a Gamma distribution is better from this plot, particularly for the 25 km distribution, which appears bimodal. Maybe a fitted lognormal distribution in panel B as a reference would make this clearer?

**Figure 7**: More information on this plot is needed. Is this a simulation at each scattering angle shown, or an average over many orbits? Is this using real data or simulated? You mention the scene reflectivity (and presumably zenith angle) is an important factor in the sensitivity, but that value is not mentioned here.

**Figures 6-7**: These figures nicely relate the gamma parameters to more physical quantities and the impact of a particular change ($\alpha, \beta \pm 10\%$) on the retrieval. However, I think the piece of information that is needed to interpret the results is how much the fitted gamma parameters vary in the CARMA model, and how much the phase function varies over this range.

**Page 11, Line 5-8**: If the difference in phase function ratio and retrieved extinction ratios is due to multiple scattering, wouldn't the smearing effect be more pronounced at 20.5 km, rather than 25.5 km? Lower altitudes generally have a larger multi-scatter component to the signal.

**Figure 11**: I think it is important to show the wavelength relationship for other altitudes. Particularly if only the CARMA data at 20.5 km was used to generate the ASD used in the retrieval.

**Page 14, Line 7**: It should maybe be mentioned that the retrieval is performed at 675 nm, so the residual must (presumably) be zero at this wavelength for both methods?

**Page 15, Line 13-14**: From Figures 11/12, the spectral dependence seems to be affected for the entire Northern hemisphere. From Figure 5, this could range from about 60-120°, please define "small $\Theta$".

**3   Technical Comments**

**Page 3, Line 20**: Seems odd to start a paragraph with an equation, should it come after line 12?

**Page 4, Line 21**: At 20 and 25 km altitudes?

**Page 6, Line 20**: CARAM to CARMA

**Page 6, Line 21**: GD distribution = Gamma Distribution distribution?

**Page 11, Line 5**: duo to due

---

## Referee Comment (RC2) · Anonymous Referee #2 · 14 Mar 2018

This paper investigates the impact of the assumed aerosol size distribution (ASD) used in the retrieval of information from radiances measured using limb scattering, and in particular here, OMPS. The topic is important since most current satellite-borne measurements of stratospheric aerosol use this technique, OSIRIS and SCIAMACHY in addition to OMPS, and since the details of how the measurements are used to retrieve the quantities of interest are not well known outside of the retrieval community. Unfortunately, although the topic of this paper is important, it fails in many aspects.

The paper begins with an unfair comparison between the currently assumed ASD for OMPS retrievals and a model distribution from a different altitude, location, and altitude.

[Figure]

I can think of many reasons why the currently assumed ASD was a poor choice. But the fact that the assumed ASD differs from the modeled ASDs, which were never intended to mimic the currently assumed ASD, is not one of them. Yet that is what the paper does and delves into details about how these distributions differ, which patently makes no sense. Of course they are different. It is fine to use a different ASD to analyze the OMPS measurements and to show how that impacts the results, but don't begin by claiming some a priori improvement in the ASD because the new and assumed ASD differ. More detail is provided below in comment 4.7-7.9.

The paper would benefit from a more complete explanation of how aerosol extinction is derived from the OMPS measured radiances, leading to the variations seen in Figure 9 by altitude and latitude. A clear simple explanation of this is missing. Here is my understanding. Is it correct?

As a function of latitude the OMPS measurement comes from a specific angle based on the solar-satellite geometry, Figure 5. The assumed ASD is used to calculate the phase function, but for any one measurement only a small piece of the phase function is important, and which piece is indicated by Figure 4. Now the radiative transfer equation is solved, with, for the aerosol, the only adjustable parameter the aerosol total number concentration, at least that piece of the number concentration which influences scattering at the wavelengths measured. Thus the OMPS radiances are used to determine the number concentration which has to be used with the normalized ASD to finally calculate aerosol extinction using Mie theory. If this is correct, something along these lines needs to be added to the paper. If it is not correct, a more correct explanation needs to be added. Presently, the authors are asking a lot of readers not intimately knowledgeable about the fine details of analyzing limb scattering measurements.

The paper lacks clarifying details. Here are some issues with further explanation below.

When extinction or extinction ratios are calculated what wavelengths are used, Figures 7, 8, 9, 10? Figure captions lack information. It is not clear how units are included in

an ASD described by a Gamma distribution. What is meant by more Rayleigh-like? The explanation of why the ASD extinction ratio has a correlation with reflectivity in the southern hemisphere is insufficient.

Finally the first statement, 15.2-5, in the conclusions section is not correct nor acceptable.

Where has it been shown, ". . . that P($\theta$) is very sensitive to the assumed aerosol particle number density near a particle radius of 0.1 micron"? Which figures? Where is the phase function shown as a function of particle size, or how this dependence figures into the impact on calculated radiances and extinctions? What the authors have shown is that if an assumed ASD, based on model results (for a time period, altitude, and location different than the previously assumed ASD), is used in place of the previously assumed ASD, then there will be differences in the calculated radiances and extinctions. But to then extend this difference to a condemnation of in situ measurements for poorly characterizing particles near 0.1 $\mu$m does not follow. This last statement may be true or false, but the results here, which use one ASD from in situ measurements, ignoring the 1000s-10,000s of other in situ ASD measurements available from aircraft and balloon, provide no answer. What the results here do show is that if an ASD from measurements two months after Pinatubo, at 16.5 km, are used for the assumed OMPS ASD, then the results are not as good as results using a more climatological ASD from 20 and 25 km. But this conclusion seems on face value to be quite obvious and not requiring all this work to prove. It seems what this paper is really about is the sensitivity of spectral extinction and radiances of the OMPS limb profiler to the assumed ASD. This can be done by choosing two quite widely divergent ASDs to compare, which is more or less what is done here, but without stating this fact and reading too much into the differences in ASD.

Here are more specific questions, comments, and corrections for this paper by page and line number.

[Figure]

3.9-11. This is an odd choice of an aerosol size distribution (ASD) to characterize the stratosphere, since this ASD would have been heavily influenced by the eruption of Pinatubo in June 1991. At least some words should be added to justify the choice. I am confident that there are many other ER-2 ASDs available in a less perturbed stratosphere. Note the values of Angstrom exponent (AE) and extinction in Figure 1 for the time period selected for the ASD in Table 1. Thus imposing a restriction of AE on the ER-2 measurements also seems artificial, and not reflective of the measurements or the time period.

Table 1. What is fc?

Figure 2. Needs more explanation and a better figure caption. What do the lines in Fig. 2a) represent? Are these just connecting the dots? Why not show the differential Gamma distribution (GD) for comparison to the model results? Which of the distributions is shown in Fig. 2b), or is a single GD with a single set of $\alpha$ and $\beta$ used for both altitudes? If the latter is the case then do the distributions only differ by a total number concentration? In line with the disparity between the time period and altitudes chosen for the ASDs to compare, how would the Pueschel ASD appear in Figure 2b), also normalized to 1 at the smallest sizes? Why are there so many fewer model points in red in Figure 2b) compared to the model points in Figure 2a)?

Eq. 2. I don't understand the units in this equation? The n(r) suggests a differential ASD in standard usage. The only units on the right appear in $r\hat{}(\alpha-1)$ and $\beta\hat{}\alpha$, so the units are m$\hat{}$-1, which is correct for a normalized differential distribution, but then there must be an No appearing in Eq. 2. In short how does the GD provide a number concentration (m$\hat{}$-3) as implied in Eq. 5 or a differential number concentration (m$\hat{}$-4) as implied in Eq. 2?

Eq. 3. What is the upper limit of the integral? There is a problem with the equation editor, so that it looks like the integral is from 0 to 0.

6.5. This is a nice result to see.

[Figure]

6.11. I believe the authors mean, . . . using a Levenberg-Marquardt nonlinear least squares regression algorithm, rather than "by".

6.20-21. "CARAM data" and "GD distribution"?

4.9-7.9 and Figures 3 and 4. This entire discussion beginning with the introduction of the CARMA modeled ASD needs to be changed. What the authors have shown with the present discussion is that the CARMA modeled ASD does not agree with Pueschel et al.'s measured ASD. Why should they be similar? Pueschel's measurements were made at 16.5 km in August 1991 at 36 N and 121 W, approximately 2 months following Pinatubo. The CARMA results are from a three year summertime climatology at 20 and 25 km at 41 N and 105 W. Of course these two ASDs are different. The text here is comparing apples and oranges and claiming they are different. Well yes they are different, but we knew that. If the authors really want to make the case that GD fits to CARMA are better than lognormal fits to measurements, then let them either compare CARMA with the dates and altitudes of the Pueschel results, or compare CARMA with measurements over Laramie, which they claim are the reasons they produced CARMA results at that position. Or better yet just compare GD and lognormal fits to the same CARMA data.

7.9-7.10. More Rayleigh-like? What is the basis for this statement? A Rayleigh phase functions varies from 0.07 to 0.11. Pueschel's phase function varies from 0.83-0.02 and is closer to either of the CARMA phase functions than Rayleigh.

7.14-15. This seems a little surprising since extinction is the loss of light in the forward direction.

9.5-6. Why would we expect that a single quantity, AE, would be enough information to determine two fitting parameters?

Figure 7 and 10.1-3. What is meant by extinction ratio and phase function ratio?

11.1. I assume the ratios of aerosol extinctions are the ratio of 525/1020 nm, but this

should be stated somewhere and it would make sense to include this information on the figures, or at least in the figure captions. Or is this the ratio of extinctions at some unspecified wavelength for the two ASDs? Or is this a ratio of ratios? Some clarification is needed.

11.3 and Figure 8. How is the ratio of phase functions calculated? How can there be a single phase function by latitude, since the phase function is angularly dependent? Since a ratio is shown why use the inverse? The phase function ratios in Figure 7 are all near 1. Why now the shift from 20/25 to 20.5/25.5 km?

Now I think I understand what is being done. Perhaps Fig. 5 could be modified to add a second panel to show that for any latitude there is only a single value, or perhaps a small angular range, of the phase function that applies, depending on the season. Then when the ratio of phase functions are discussed it will be clear what ratios are being used. It would be very helpful to show the variation of phase function with latitude, conflating Figures 4 and 5, for the two ASDs.

11.7. Now multiple scattering is brought in which has not so far not been mentioned. This seems rather cavalier, since no further mention is made of multiple scattering. Duo?

Figure 9. Again! What wavelength extinctions are being shown? The limitations on the piece of the phase function used by latitude for any measurement helps immensely to explain why there is very little variation in the ASD extinction ratio between 40 and 80 N in Figure 9.

12.7. Right, once the extinction calculation is understood, it is clear that a lower value of the phase function will lead to larger number concentrations for the same limb scattering measurement, thus larger extinction.

12.9-20 and Figure 10. Why is the ASD extinction ratio correlated with reflectivity in the southern hemisphere? The authors do not explain this, they sort of imply that

reflectivity has a larger variation in the southern hemisphere, but this is not the case. The reflectivity variation is similar in both hemispheres. It really comes down, again, to the conflation of Figures 4 and 5, illustrating how the southern hemisphere is so much more sensitive to the choice of ASD, than the northern hemisphere, due to the larger differences in backscatter compared to forward scatter.

Fig. 11 and 14.1-6. Panels are also shown for 745 and 869 nm. Why aren't these in the figure caption and mentioned in the text? What is the explanation for the larger residuals for the Pueschel ASD in the northern hemisphere? This seems surprising given the tight extinction ratio, Figure 9, and the similarity of the phase functions, Figure 4, for the two ASDs in the northern hemisphere, and since ASI is proportional to $E^*P(\theta)$.

---

## Referee Comment (RC3) · Anonymous Referee #3 · 20 Mar 2018

The paper titled "Impact of aerosol size distribution on extinction and spectral dependence of radiances measured by the OMPS Limb profiler instrument" by "Zhong Chen1, Pawan K. Bhartia2, Robert Loughman3, and Peter Colarco2" examines the impact of aerosol size distribution from CARMA on OMPS extinction. While the paper is well written, I think there is still scope for improvement particularly the way the paper is emphasized on the impact of aerosol size distribution on deriving extinction. I also see that the topic is of great importance to the stratospheric aerosol community as improvement on OMPS aerosol retrieval will be a very good addition to other limb scatter measurements such as OSIRIS, and SCIAMACHY.

[Figure]

Although, I understand it's a comparison between assumed aerosol size distribution in version 1 of OMPS aerosol retrieval and ASDs from CARMA, I think the comparison here is probably not correct as model has its own pros and cons when it comes to aerosol size distributions. While I have no issues in using modeled ASDs in this study, I would like see a comparison between modeled ASDs/extinction and available observations (e.g. balloon measurements). I also see an issue here as the ASD derived from CARMA used in this study is from a period 1990-1993 which is when the stratosphere is highly volcanically affected due to Mount Pinatubo. I believe that using these ASDs for the study here will have a clear impact on extinction as aerosol size distributions vary from these two periods.

From the paper, I understand that for version 1, the ASD is derived from aircraft measurements. I wonder why only this aircraft measurement was used while there were many other aircraft/in situ balloon measurements were available. For example, for OSIRIS/SCIAMACHY retrieval balloon in situ measurements during low volcanic/background period were used for ASDs. My concern is that ASDs from a large volcanically perturbed time period may not be a correct assumption as there were many moderate volcanic event post-SAGEII era (after 2005). This is what was done in version 1 of OMPS retrieval and similar volcanically influenced ASDs are used in the model simulated ASDs here if I understand it correctly.

The paper fits within the scope of the journal. Although, the paper is reasonably well written, I find that there is a lot of scope for improvement. So, I recommend some revisions below:

Major comments:

Page 3 L10: Why are ER-2 measurements in August 1991 was used while there were many other ER-2 measurements/balloon measurements during moderate/background periods were available which will be more realistic in terms of OMPS period of measurements?

[Figure]

Page 4 L10: I am not an expert in running models but I am not sure how the simulations were done here? The sentence reads as "no explosive eruptions were used for the precursor emission but then in line 19 it reads as the simulation was done for the period 1990-1993 which includes Mount Pinatubo time period. I think it would be helpful for readers if you could explain this a little bit more in detail.

As from the model simulations, I believe the simulations were made using prescribed SSTs for the period 1990-1993. My concern here is that a highly volcanically influenced ASDs are used here as this may not be a correct way of representing ASDs for the stratosphere for the OMPS measurement time period which includes many moderate eruptions. May be, it is more realistic if the simulation was done with same prescribed SSTs for the post-Pinatubo period (post 2005) to represent more of moderate volcanic eruptions.

Page 5 L 4-10: How does OPC's compare to these distributions? I would like to see a comparison here. Although, gamma distribution in this case may be a better representation, I still believe that lognormal distribution is the best possible representation of stratospheric aerosols which I think would fit very well to the observations.

I would like to see a sensitive analysis to Gamma distribution and lognormal distribution and compare them with actual measurements available on an altitude basis. I would like to see how these distributions differ particularly near tropopause region and higher up. Probably showing a comparison at different altitudes may help understand the observations better.

The other possible way to compare your results is to compare CARMA ASDs with OPC measurements from Deshler et al., 2003 as balloon measurements have higher vertical resolution than aircraft measurements which will give us an idea how CARMA compares with the observed size distribution. I believe this is an important point to make as authors are testing a new ASD from a model in this study and this point should be addressed.

Page 7 L10: I am not sure what this means? "We find that the key difference between the two ASDs is that the Pueschel distribution has larger dN/dlogr values at 0.1 micron, which causes the derived aerosol scattering phase function P( ), shown in Figure 4, to be more "Rayleigh-like" at large single scattering angle , i.e., closer to the Rayleigh P("

Page 10: It may help the reader if authors could explain as how the extinction is computed and at what wavelengths the extinctions are calculated.

Page 11 Figure 9: How does it look like in the lower stratosphere. This is where the main issue of all limb scatter measurements lies. I would like to see a similar plot for lower altitudes say 18, 16, or 13 km. If you could use a tropopause height climatology and use the above altitudes to do a similar plot, I expect to see some data showing up at 13 km for higher latitudes where I think limb scatter measurements have issues. What wavelength is extinction in Figure 9 calculated at?

Page 13 Figure 11: The figure says ASI's are computed at three different wavelengths. Are these wavelengths just used for ASI's or are these used for computing extinction as well (for example in Figure 9)?

Page 15 L15-20: The OSIRIS data are in reasonably good agreement with SAGEII (Rieger et al., 2015) except in the lower stratosphere at higher latitudes. I would like to see how CARMA ASD's derived extinction compare to OSIRIS. I understand it may be out of the scope of this paper but it would definitely help the stratospheric aerosol community as I believe OMPS measurements are valuable which may help fill the gap between SAGEII and SAGEIII-ISS in addition to OSIRIS/SCIAMACHY/CALIPSO.

---

## Author Comment (AC1) · 6 Apr 2018

**Reply to Reviewer 1**
**Z. Chen et al.**
zhong.chen@ssaihq.com

The valuable comments by Reviewer 1 are greatly appreciated. Our replies to the Reviewer 1 comments are given below.

*1. General Considerations: This article discusses the retrieval of stratospheric aerosol extinction profiles using the OMPS-LP measurements. The authors use a gamma particle size distribution derived from the CARMA model instead of the standard lognormal assumption, and it is found that this helps to improve the spectral response of the modelled signal at 20.5 km. The approach is a novel one and valuable to the limb scattering aerosol retrieval community. The writing is concise, and the material is generally well explained; after the following minor edits I recommend publication.*
**Reply:** We thank Reviewer 1 for the positive comments.

*2. Specific Comments: Some additional information on how and why the CARMA ASD was chosen would be beneficial. It is not clear what sampling is used to derive the parameters in Table 2. What years are the June-July-August data from, what altitudes are used, etc. Is a single GD chosen due to retrieval requirements, or other reasons? Why is sampling near Laramie important if the balloon data is not compared against?*
**Reply**: We add 'at 20km' in Table 2. A single GD for 20km is chosen due to retrieval requirements. We add explanatory text: 'In the retrieval algorithm, we assume that the size distribution is height independent, so that one GD at 20 km is used to represent the aerosol size distribution at all heights.'

As indicated, we are leveraging some CARMA simulations that were performed as part of a Pinatubo-focused study, so all simulations are for the period 1990 – 1993. The June-July-August data used is the average of that season over those 4 years. In this paper we are using only CARMA Wyoming data because we did validate offline that the ASD there was well reflected in the balloon data (from non-volcanic periods), (Kovilakam and Deshler, 2015).

*Figure 1: Usually the majority of the increase in extinction is attributed to Ruang/ Reventador in late 2002 (from the figure it appears the increase starts before 2003) and Manam in 2005 (eg. Vernier et al, 2011). Is there a reason the increase is attributed to Anatahan here?*
**Reply:** Thank you for pointing this. We updated the caption, and changed the text to 'Ruang/Reventador in late 2002 and Manam in 2005'.

*Figure 2: Why is only the 20km altitude shown in panel B? Also, why are only select CARMA radii used as comparison points (red dots) in panel B and not all of them?*
**Reply:** Because ASD at 20 km is used in the retrieval (see Reply to Specific Comments above). We add text: 'The cumulative CARMA data (circles) are chosen in consisting with the OPC in situ measurements which has a gap ranging from 0.01 μm to 0.1 μm (Kovilakam and Deshler, 2015)'

***Page 5, Line 15-16****: It is not clear that a Gamma distribution is better from this plot, Particularly for the 25 km distribution, which appears bimodal. May be a fitted lognormal distribution in panel B as a reference would make this clearer?*

**Reply:** We agree with you that lognormal distribution would fit very well to the observations. In our case, however, a Gamma size distribution (GD) represents a significantly better fit to the CARMA data than a unimodal normal distribution (UD) or a bimodal lognormal size distribution (BD), and our comparison results between the calculated and the observed ASIs suggest that Gamma-CARMA ASD is suitable for OMPS/LP measurements. The following two figures show the fitting results.

[Figure]

**Figure R1.** Gamma size distribution (GD), unimodal normal distribution (UD) and bimodal lognormal size distribution (BD) fits to the cumulative number density (N>r) for 20 km (a) and 25km (b). For the purpose of comparison, the cumulative CARMA data between 0.02 - 0.1 μm are also shown.

To make this clearer, we add a GD fit in Figure 2b for 25 km as you suggested. You can find that the GD with just 2 parameters fits CARMA data at 25km also well.

***Figure 7****: More information on this plot is needed. Is this a simulation at each scattering angle shown, or an average over many orbits? Is this using real data or simulated? You mention the scene reflectivity (and presumably zenith angle) is an important factor in the sensitivity, but that value is not mentioned here.*

**Reply:** Figure 7a shows the simulated phase function changes at each scattering angle and Figure 7b shows the effect of the phase function changes on the retrieved extinction using OMPS/LP measurements for a single orbit on September 12, 2016. We have added the information on this Figure. The effect of reflectivity is also mentioned hear.

***Figures 6-7****: These figures nicely relate the gamma parameters to more physical quantities and the impact of a particular change (α,β±10%) on the retrieval. However, I think the piece of information that is needed to interpret the results is how much the fitted*

*gamma parameters vary in the CARMA model, and how much the phase function varies over this range.*

**Reply:** We have added text 'It is apparent that the phase function is quite sensitive to β. A ±10% change in β can produce a ±10% change in the calculated aerosol phase function, whereas for a 10% change in α the percentage change of aerosol phase function is within ±3%.'

*Page 11, Line 5-8: If the difference in phase function ratio and retrieved extinction ratios is due to multiple scattering, wouldn't the smearing effect be more pronounced at 20.5 km, rather than 25.5 km? Lower altitudes generally have a larger multi-scatter component to the signal.*

**Reply:** That is correct. We have added text 'Since lower altitudes generally have a larger multi-scatter component to the signal, the smearing effect is more pronounced at 20.5 km, rather than at 25.5 km'.

*Figure 11: I think it is important to show the wavelength relationship for other altitudes. Particularly if only the CARMA data at 20.5 km was used to generate the ASD used in the retrieval.*

**Reply:** We have added a second panel in Figure 12 to show the wavelength relationship for another altitude at 25.5km.

*Page 14, Line 7: It should maybe be mentioned that the retrieval is performed at 675 nm, so the residual must (presumably) be zero at this wavelength for both methods?*

**Reply:** That is true. We have added text 'Note that our aerosol retrieval is performed at 675 nm, so the ASI residual at this wavelength are very close to zero for both methods'.

*Page 15, Line 13-14: From Figures 11/12, the spectral dependence seems to be affected for the entire Northern hemisphere. From Figure 5, this could range from about 60-120°, please define "small Θ".*

**Reply:** That is correct. We have changed 'except at small Θ' to 'for the southern hemisphere'.

*3 Technical Comments*
*Page 3, Line 20: Seems odd to start a paragraph with an equation, should it come after line 12?*
**Reply:** Fixed.

*Page 4, Line 21: At 20 and 25 km altitudes?*
**Reply:** The '20 km altitude' has been changed to '20 and 25 km altitudes'.

*Page 6, Line 20: CARAM to CARMA*
**Reply:** Fixed.

*Page 6, Line 21: GD distribution = Gamma Distribution distribution?*
**Reply:** The 'GD distribution' has been changed to 'Gamma distribution'.

***Page 11, Line 5****: duo to due*
**Reply:** Done.

---

## Author Comment (AC2) · 6 Apr 2018

**Reply to Reviewer 2**
**Z. Chen et al.**
zhong.chen@ssaihq.com

We thank Reviewer 2 for reading our paper in detail and providing so much comments and suggestions. Point-by-point responses are numbered in the same order as Reviewer 2 was given in the review:

*Comment 1: This paper investigates the impact of the assumed aerosol size distribution (ASD) used in the retrieval of information from radiances measured using limb scattering, and in particular here, OMPS. The topic is important since most current satellite-borne measurements of stratospheric aerosol use this technique, OSIRIS and SCIAMACHY in addition to OMPS, and since the details of how the measurements are used to retrieve the quantities of interest are not well known outside of the retrieval community. Unfortunately, although the topic of this paper is important, it fails in many aspects. The paper begins with an unfair comparison between the currently assumed ASD for OMPS retrievals and a model distribution from a different altitude, location, and altitude. I can think of many reasons why the currently assumed ASD was a poor choice. But the fact that the assumed ASD differs from the modeled ASDs, which were never intended to mimic the currently assumed ASD, is not one of them. Yet that is what the paper does and delves into details about how these distributions differ, which patently makes no sense. Of course they are different. It is fine to use a different ASD to analyze the OMPS measurements and to show how that impacts the results, but don't begin by claiming some a priori improvement in the ASD because the new and assumed ASD differ. More detail is provided below in comment 4.7-7.9.*

**Reply:** We agree that the two ASDs are different in time, altitude, location, and altitude. However, one should consider these conditions in interpreting the comparison. Naturally, it is fairly common practice to compare the two ASDs that are used in the retrieval algorithm to retrieve aerosol extinction using OMPS/LP measurements under the same geophysical conditions to see how a change in ASD affects the retrieved extinction values and to determine which ASD is better. This is done by comparing the calculated ASIs from the two ASDs with the measured ASIs. These types of comparisons helped us to improve the retrieval by choosing a correct ASD.

*Comment 2: The paper would benefit from a more complete explanation of how aerosol extinction is derived from the OMPS measured radiances, leading to the variations seen in Figure 9 by altitude and latitude. A clear simple explanation of this is missing. Here is my understanding. Is it correct? As a function of latitude the OMPS measurement comes from a specific angle based on the solar-satellite geometry, Figure 5. The assumed ASD is used to calculate the phase function, but for any one measurement only a small piece of the phase function is important, and which piece is indicated by Figure 4. Now the radiative transfer equation is solved, with, for the aerosol, the only adjustable parameter the aerosol total number concentration, at least that piece of the number concentration which influences scattering at the wavelengths measured. Thus the OMPS radiances are used to determine the number concentration which has to be used with the normalized*

*ASD to finally calculate aerosol extinction using Mie theory. If this is correct, something along these lines needs to be added to the paper. If it is not correct, a more correct explanation needs to be added. Presently, the authors are asking a lot of readers not intimately knowledgeable about the fine details of analyzing limb scattering measurements.*

**Reply:** The following explanatory text was now added:
'We retrieve aerosol extinction profiles at 675 nm from OMPS/LP radiance measurements. We first calculate the cross-section and phase function from the resulting Gamma aerosol size distribution using Mie theory, then run the radiative transfer model within the aerosol retrieval code with the new cross-section and phase function to determine how the OMPS/LP aerosol retrieval changes with this new ASD relative to Pueschel ASD in the V1 (Loughman et al., 2018).'

***Comment 3:*** *The paper lacks clarifying details. Here are some issues with further explanation below. When extinction or extinction ratios are calculated what wavelengths are used, Figures 7, 8, 9, 10? Figure captions lack information. It is not clear how units are included in an ASD described by a Gamma distribution. What is meant by more Rayleigh-like? The explanation of why the ASD extinction ratio has a correlation with reflectivity in the southern hemisphere is insufficient.*

**Reply:** OMPS aerosol extinctions retrievals are retrieved using wavelength at 675 nm, as indicated. We have added a note to the caption and text reminding the reader that the OMPS/LP aerosol retrieval is performed at 675 nm (see Reply to Comment 2 above). Eq. 2 presented a normalized differential size distribution. We now added $N_0$ with text "*n(r)* is the size distribution function ($cm^{-3}\mu m^{-1}$), $N_0$ is the total number density of aerosols ($cm^{-3}$)'. We also added the units of $\beta$ ($1/\mu m$) here and in the Table 2. More Rayleigh-like means that comparing with CARMA phase function, Pueschel phase function is closer to Rayleigh phase function at larger $\Theta$. Regarding the comment on the correlation between extinction and reflectivity in the southern hemisphere, it shows that this relationship is rather complex. However, we accept the comment made by Reviewer 2 (see Reply to Comment 19 below).

***Comment 4:*** *Finally the first statement, 15.2-5, in the conclusions section is not correct nor acceptable. Where has it been shown, "… that P(Θ) is very sensitive to the assumed aerosol particle number density near a particle radius of 0.1 micron"? Which figures? Where is the phase function shown as a function of particle size, or how this dependence figures into the impact on calculated radiances and extinctions? What the authors have shown is that if an assumed ASD, based on model results (for a time period, altitude, and location different than the previously assumed ASD), is used in place of the previously assumed ASD, then there will be differences in the calculated radiances and extinctions. But to then extend this difference to a condemnation of in situ measurements for poorly characterizing particles near 0.1 μm does not follow. This last statement may be true or false, but the results here, which use one ASD from in situ measurements, ignoring the 1000s-10,000s of other in situ ASD measurements available from aircraft and balloon, provide no answer. What the results here do show is that if an ASD from measurements*

*two months after Pinatubo, at 16.5 km, are used for the assumed OMPS ASD, then the results are not as good as results using a more climatological ASD from 20 and 25 km. But this conclusion seems on face value to be quite obvious and not requiring all this work to prove. It seems what this paper is really about is the sensitivity of spectral extinction and radiances of the OMPS limb profiler to the assumed ASD. This can be done by choosing two quite widely divergent ASDs to compare, which is more or less what is done here, but without stating this fact and reading too much into the differences in ASD.*

**Reply:** We stand by this statement. Our findings include that data near a particle radius of 0.1 $\mu$m is impotent for deriving accurate phase function, especially for using a bimodal lognormal size distribution. As an example, Figure 3 compares the two ASDs and highlights the differences at 0.1 $\mu$m and radii greater than 0.3 $\mu$m. Figure 4 shows the phase function as a function of scatter angle for different ASDs which have different peak values near 0.1 $\mu$m shown in Figure 3. We have added an appendix, and Figures A1-A2 in it speak to the point of this comment.

We don't think looking at the individual contributions is as helpful here. We expect a broad range of particle sizes, and Figures A1-A2 in Appendix A speak to the concern that motivated this comment.

Figure R2. Mie phase functions for different values of the size parameter $\chi$ derived with a refractive index of 1.33. Observe the increasing asymmetry and complexity of the phase functions with increasing $\chi$ (Petty, 2006).

We definitely do not condemn in situ measurements for poorly characterizing particles near 0.1 $\mu$m does not follow. However, almost all OPC measurements have limitations below 0.1 µm such that the aerosol size distribution from 0.01 – 0.1 µm is poorly measured. In fact, the Wyoming in situ data have been updated recently and the minimum size measured is now claimed to be 0.094 $\mu$m (Deshler, private communication, 2018). We now added text and Appendix A below in support of this point: 'Additionally, most OPC measurements have limitations below 0.1 µm such that the aerosol size distribution from 0.01 - 0.1 µm is poorly measured. The lack of information in the OPC data gap region would result in uncertainty in calculating phase function (see Appendix A).'

Regarding the comment on the ASD and ASI comparisons, please see Reply to Comment 1. We would emphasize that these types of comparisons yield scientifically important insights. What we have shown is that ASIs from the Gamma/CARMA ASD agree better with the OMPD/LP measurements than the ASIs from the Pueschel ASD, not "*the CARMA modeled ASD does not agree with Pueschel et al.'s measured ASD*", as repeated by Referee 2. Based on this comment of the reviewer we realized, that there might be a misunderstanding of the general approach of our study, or it is possible to mix ASI up with ASD.

***Comment 5:*** *3.9-11. This is an odd choice of an aerosol size distribution (ASD) to characterize the stratosphere, since this ASD would have been heavily influenced by the eruption of Pinatubo in June 1991. At least some words should be added to justify the*

*choice. I am confident that there are many other ER-2 ASDs available in a less perturbed stratosphere. Note the values of Angstrom exponent (AE) and extinction in Figure 1 for the time period selected for the ASD in Table 1. Thus imposing a restriction of AE on the ER-2 measurements also seems artificial, and not reflective of the measurements or the time period.*

**Reply:** We agree that there are many other bimodal lognormal size distributions resulting from ER-2 samples. We specifically wanted a bi-modal distribution with a "fine mode" + a "coarse mode", and tuned the coarse mode fraction to give the desired Angstrom coefficient = 2 (which of course nearly made the coarse mode vanish). Now we've tried something else that seems to explain the ASI better. We added text "Our main motivation for using this Pueschel bimodal size distribution arose from the existing OPC dataset, which generally features a bimodal size distribution at the altitudes where the stratospheric aerosol extinction is greatest. But the problem of how to specify this more complex distribution is a serious concern. Our initial hope was that requiring the resulting Angstrom exponent to = 2 would minimize the importance of the 5 size parameter settings, but that is unfortunately not true in all cases."

*Comment 6: Table 1. What is fc?*

**Reply:** We added text to Table 1 "fc is the coarse mode fraction, which is the ratio of the number of particles of the coarse mode to the total number of particles for a bimodal lognormal distribution (Loughman et al., 2018)."

*Comment 7: Figure 2. Needs more explanation and a better figure caption. What do the lines in Fig. 2a) represent? Are these just connecting the dots? Why not show the differential Gamma distribution (GD) for comparison to the model results? Which of the distributions is shown in Fig. 2b), or is a single GD with a single set of α and β used for both altitudes? If the latter is the case then do the distributions only differ by a total number concentration? In line with the disparity between the time period and altitudes chosen for the ASDs to compare, how would the Pueschel ASD appear in Figure 2b), also normalized to 1 at the smallest sizes? Why are there so many fewer model points in red in Figure 2b) compared to the model points in Figure 2a)?*

**Reply:** We have added 'The lines in (a) are just connecting the circles.' Since we use ASD at 20km in the retrieval, Fig. 2b showed a Gamma function fit to the cumulative number density just for 20km only (for the purpose of comparison, we now added a GD fit for 25km as Reviewer 1 suggested). We do not use a single GD for both altitudes. So the latter is not the case. We added text: 'The cumulative CARMA data (circles) are chosen in consisting with the OPC in situ measurements which has a gap ranging from 0.01 µm to 0.1 µm (Kovilakam and Deshler, 2015).'

*Comment 8: Eq. 2. I don't understand the units in this equation? The n(r) suggests a differential ASD in standard usage. The only units on the right appear in $r^{(\alpha-1)}$ and $\beta^{\alpha}$, so the units are m^-1, which is correct for a normalized differential distribution, but then there must be an No appearing in Eq. 2. In short how does the GD provide a number*

*concentration (mˆ-3) as implied in Eq. 5 or a differential number concentration (mˆ-4) as implied in Eq. 2?*

**Reply:** Eq. 2 presented a normalized differential size distribution. We now added $N_0$ with text "$n(r)$ is the size distribution function ($cm^{-3}\mu m^{-1}$), $N_0$ is the total number density of aerosols ($cm^{-3}$)'. We also added the units of $\beta$ ($1/\mu m$) here and in the Table 2. Please see Reply to Comment 3 above.

***Comment 9:*** *Eq. 3. What is the upper limit of the integral? There is a problem with the equation editor, so that it looks like the integral is from 0 to 0.*

**Reply:** The document that we see does the integral in equation 3 with respect to r from 0 to $\infty$, not from 0 to 0. In the pdf version on the website, the same thing appears. Now we replaced $\int_0^\infty$ with $\int_0^\infty$ . Is it ok?

***Comment 10:*** *6.11. I believe the authors mean, ... using a Levenberg-Marquardt nonlinear least squares regression algorithm, rather than "by".*

**Reply:** We replaced 'by' with 'using'.

***Comment 11:*** *6.20-21. "CARAM data" and "GD distribution"?*
**Reply:** Reviewer 1 also noted this. The "GD distribution" has been changed to "Gamma distribution ".

***Comment 12:*** *4.9-7.9 and Figures 3 and 4. This entire discussion beginning with the introduction of the CARMA modeled ASD needs to be changed. What the authors have shown with the present discussion is that the CARMA modeled ASD does not agree with Pueschel et al.'s measured ASD. Why should they be similar? Pueschel's measurements were made at 16.5 km in August 1991 at 36 N and 121 W, approximately 2 months following Pinatubo. The CARMA results are from a three year summertime climatology at 20 and 25 km at 41 N and 105 W. Of course these two ASDs are different. The text here is comparing apples and oranges and claiming they are different. Well yes they are different, but we knew that. If the authors really want to make the case that GD fits to CARMA are better than lognormal fits to measurements, then let them either compare CARMA with the dates and altitudes of the Pueschel results, or compare CARMA with measurements over Laramie, which they claim are the reasons they produced CARMA results at that position. Or better yet just compare GD and lognormal fits to the same CARMA data.*

**Reply:** The comment on ASD comparison repeats the previous one. Please see Replies to Comments 1 and 4 above. Regarding comparing GD and lognormal fits to the same CARMA data, we made the following two figures as Reviewer 1 suggested.

[Figure]

**Figure R1.** Gamma size distribution (GD), unimodal normal distribution (UD) and bimodal lognormal size distribution (BD) fits to the cumulative number density (N>r) for 20 km (a) and 25km (b). For the purpose of comparison, the cumulative CARMA data between 0.02 - 0.1 μm are also shown.

**Comment 12:** *7.9-7.10. More Rayleigh-like? What is the basis for this statement? A Rayleigh phase functions varies from 0.07 to 0.11. Pueschel's phase function varies from 0.83-0.02 and is closer to either of the CARMA phase functions than Rayleigh.*

**Reply:** The baseline for this statement is Rayleigh phase function. Please see Reply to Comment 3 above.

**Comment 13:** *9.5-6. Why would we expect that a single quantity, AE, would be enough information to determine two fitting parameters?*

**Reply:** We removed the unnecessary text 'Note that AE by itself does not provide information to determine both $\alpha$ and $\beta$, and hence $r_{eff}$'.

**Comment 14:** *Figure 7 and 10.1-3. What is meant by extinction ratio and phase function ratio? 11.1. I assume the ratios of aerosol extinctions are the ratio of 525/1020 nm, but this should be stated somewhere and it would make sense to include this information on the figures, or at least in the figure captions. Or is this the ratio of extinctions at some unspecified wavelength for the two ASDs? Or is this a ratio of ratios? Some clarification is needed.*

*11.3 and Figure 8. How is the ratio of phase functions calculated? How can there be a single phase function by latitude, since the phase function is angularly dependent? Since a ratio is shown why use the inverse? The phase function ratios in Figure 7 are all near 1. Why now the shift from 20/25 to 20.5/25.5 km?*

*Now I think I understand what is being done. Perhaps Fig. 5 could be modified to add a second panel to show that for any latitude there is only a single value, or perhaps a small angular range, of the phase function that applies, depending on the season. Then when the ratio of phase functions are discussed it will be clear what ratios are being used. It would be very helpful to show the variation of phase function with latitude, conflating Figures 4 and 5, for the two ASDs.*

**Reply:** There might be a misunderstanding about the ratios. Figure 7 (and Figure 6) relate the Gamma parameters to more physical quantities and the impact of a particular change (α,β±10%) on the retrieval. So the extinction ratio and phase function ratio shown in Figure 7 are the ratios of the CARMA perturbations that result from adjusting Gamma parameters by ±10% to the CARMA baseline model that is calculated using the fitted Gamma parameters (α=1.8, β=20), while Figures 8 and 9 show the impact of the two ASDs (CARMA and Pueschel) on the retrieved extinction. The labels and legends in Figures 8 and 9 clearly represent the ratio of CARMA to Pueschel. For the purpose of comparison, we use the inversed phase function ratio in Figure 8 and 9 since the change in phase function and the change in extinction are roughly anti-correlated (see Figure 7). The shift from 20/25 to 20.5/25.5 km due to we calculate P(Θ) using the assumed ASD at 20 km and we retrieve extinction using OMPS measurements at 20.5 and 25.5 km. We show phase function as a function of latitude based on the relationships shown in Figures 4 and 5. Please also note that in Figure 8, we show daily zonal mean data in $5^{\circ}$ latitude bands.

**Comment 15:** *11.7. Now multiple scattering is brought in which has not so far not been mentioned. This seems rather cavalier, since no further mention is made of multiple scattering.*

**Reply:** The review states earlier that "only a small piece of the phase function is important" (**Comment 2**). This only makes sense if multiple scattering (MS) plays a small role. And our Figure 10 clearly shows how the phase function influence declines as ρ increases (adding more diffuse radiation to the atmosphere). Please refer Loughman et al. (2018) for details.

**Comment 16:** *Duo?*
**Reply:** Reviewer 1 also noted this typo. It is fixed.

**Comment 17:** *Figure 9. Again! What wavelength extinctions are being shown? The limitations on the piece of the phase function used by latitude for any measurement helps immensely to explain why there is very little variation in the ASD extinction ratio between 40 and 80 N in Figure 9.*

**Reply:** The OMPS/LP aerosol retrieval is performed using wavelength at 675 nm (see Replies to Comments 2 and 3). Your explanation may be correct.

**Comment 18:** *12.7. Right, once the extinction calculation is understood, it is clear that a lower value of the phase function will lead to larger number concentrations for the same limb scattering measurement, thus larger extinction.*

**Reply:** That is correct.

**Comment 19:** *12.9-20 and Figure 10. Why is the ASD extinction ratio correlated with reflectivity in the southern hemisphere? The authors do not explain this, they sort of*

*imply that reflectivity has a larger variation in the southern hemisphere, but this is not the case. The reflectivity variation is similar in both hemispheres. It really comes down, again, to the conflation of Figures 4 and 5, illustrating how the southern hemisphere is so much more sensitive to the choice of ASD, than the northern hemisphere, due to the larger differences in backscatter compared to forward scatter.*

**Reply:** It is true that the reflectivity variation is similar in both hemispheres. We agree with you and added text: 'The conflation of Figures 4 and 5 illustrates that how the southern hemisphere is so much more sensitive to the choice of ASD, than the northern hemisphere, due to the larger differences in back scatter compared to forward scatter'.

*Comment 20: Fig. 11 and 14.1-6. Panels are also shown for 745 and 869 nm. Why aren't these in the figure caption and mentioned in the text? What is the explanation for the larger residuals for the Pueschel ASD in the northern hemisphere? This seems surprising given the tight extinction ratio, Figure 9, and the similarity of the phase functions, Figure 4, for the two ASDs in the northern hemisphere, and since ASI is proportional to E\*P(Θ).*

**Reply:** Thanks for pointing this out. We now added "745 and 869 nm" to the caption and text. We also added explanatory text 'The larger residuals for the Pueschel ASD in the northern hemisphere suggest that CARMA ASD is better than Pueschel ASD for OMPS/LP measurements'.

**Appendix A. Fitting Aerosol Size Distributions to OPC data**

One of the longest and most comprehensive records of local stratospheric aerosol conditions is from the University of Wyoming's optical particle counters (OPC) carried on weather balloons at Laramie, Wyoming, USA ($41°N$) at altitudes up to 30km. The instrument measures the number of aerosol particles in several size bins, ranging from 0.15 to 2 μm. In most cases, bimodal lognormal size distributions (BD) is used to fit OPC data if there are enough sizes measured. For background stratospheric conditions, however, OPC data often does not provide sufficient information for a robust BD fit. An example of this problem is illustrated in Figure A1, which shows four deferent BD fits to the same OPC concentration data of April 12, 2000 for an altitude at 20 km (Kovilakam and Deshler, 2015). Since the five parameters of a BD are interdependent, the fits were constrained by using different values of $f_c$. The fitted parameters as well as the calculated AE and $r_{eff}$ are given in Table A1.

[Figure]

**Figure A1.** Estimated bimodal lognormal cumulative distributions (a) and differential distributions (b) for nonvolcanic (20000412 implies 12 April 2000) OPC measurement at 20 km (Kovilakam and Deshler, 2015). Measurements on the left panel are shown as black dots.

**Table A1.** Four BD fits to OPC data measured at Laramie Wyoming on April 12, 2010 at 20km.

|  | ASD_1 | ASD_2 | ASD_3 | ASD_4 |
|---|---|---|---|---|
| Coarse mode fraction, $f_c$ | 0.0195 | 0.006 | 0.15 | 0.23 |
| Mode radius, $r_i$ (μm) | 0.080,0.238 | 0.075,0.280 | 0.046,0.140 | 0.040,0.120 |
| Mode Width, $\sigma_i$ | 1.45,1.25 | 1.56,1.21 | 1.45,1.43 | 1.43,1.47 |
| Angström exponent, AE | 2.45 | 2.40 | 2.40 | 2.40 |
| Effective radius, $r_{eff}$ (μm) | 0.1332 | 0.1335 | 0.1437 | 0.1470 |

The four fits have similar Angström exponents but differed from each other significantly in radius range between 0.01 μm to 0.1 μm. As a consequence, the differences between the ASDs near 0.1 μm lead to significant changes in P(Θ) as shown in Figure A2. It can be seen that P(Θ) is quite sensitive to the value of dN/dlog10r at around 0.1μm when Θ> 90$^{\circ}$. The larger this value, more Rayleigh-like the P(Θ).

[Figure]

**Figure A2.** Aerosol phase functions at 675nm as a function of single scattering angle for the four ASDs listed in Table A1.

---

## Author Comment (AC3) · 6 Apr 2018

**Reply to Reviewer 3**
**Z. Chen et al.**
zhong.chen@ssaihq.com

We appreciate the comments offered. Our replies to the Reviewer 3 major comments are given below.

*Page 3 L10: Why are ER-2 measurements in August 1991 was used while there were many other ER-2 measurements/balloon measurements during moderate/background periods were available which will be more realistic in terms of OMPS period of measurements?*

**Reply:** Reviewer 2 also noted this. We agree that there are many other ER-2 measurements/balloon measurements available. We add text 'Our main motivation for using this Pueschel bimodal size distribution arose from the existing OPC dataset, which generally features a bimodal size distribution at the altitudes where the stratospheric aerosol extinction is greatest. But the problem of how to specify this more complex distribution is a serious concern. Our initial hope was that requiring the resulting Angstrom exponent to = 2 would minimize the importance of the 5 size parameter settings, but that is unfortunately not true in all cases'.

*Page 4 L10: I am not an expert in running models but I am not sure how the simulations were done here? The sentence reads as "no explosive eruptions were used for the precursor emission but then in line 19 it reads as the simulation was done for the period 1990-1993 which includes Mount Pinatubo time period. I think it would be helpful for readers if you could explain this a little bit more in detail. As from the model simulations, I believe the simulations were made using prescribed SSTs for the period 1990-1993. My concern here is that a highly volcanically influenced ASDs are used here as this may not be a correct way of representing ASDs for the stratosphere for the OMPS measurement time period which includes many moderate eruptions. May be, it is more realistic if the simulation was done with same prescribed SSTs for the post-Pinatubo period (post 2005) to represent more of moderate volcanic eruptions.*

**Reply:** We agree that the simulations from modern period that include moderate volcanic eruptions can allay concerns over SST issues, but we think this is minor point. In fact, we are leveraging some CARMA simulations that were performed as part of a Pinatubo-focused study, so all simulations are for the period 1990 - 1993, as indicated. In the subset of simulations used in our paper we used model results that included *only* anthropogenic and non-volcanic natural sources of sulfate and precursors (the volcanoes were turned off, the natural source is from oxidation of OCS only).

*Page 5 L 4-10: How does OPC's compare to these distributions? I would like to see a comparison here. Although, gamma distribution in this case may be a better representation, I still believe that lognormal distribution is the best possible representation of stratospheric aerosols which I think would fit very well to the observations. I would like to see a sensitive analysis to Gamma distribution and*

*lognormal distribution and compare them with actual measurements available on an altitude basis. I would like to see how these distributions differ particularly near tropopause region and higher up. Probably showing a comparison at different altitudes may help understand the observations better. The other possible way to compare your results is to compare CARMA ASDs with OPC measurements from Deshler et al., 2003 as balloon measurements have higher vertical resolution than aircraft measurements which will give us an idea how CARMA compares with the observed size distribution. I believe this is an important point to make as authors are testing a new ASD from a model in this study and this point should be addressed.*

**Reply:** Reviewer 1 made similar suggestions. We agree with you that lognormal distribution would fit very well to the observations. In our case, however, a Gamma size distribution (GD) represents a significantly better fit to the CARMA data than a unimodal normal distribution (UD) or a bimodal lognormal size distribution (BD), and our comparison results between the calculated and the observed ASIs suggest that Gamma-CARMA ASD is suitable for OMPS/LP measurements. The following two figures show the fitting results.

[Figure]

**Figure R1.** Gamma size distribution (GD), unimodal normal distribution (UD) and bimodal lognormal size distribution (BD) fits to the cumulative number density (N>r) for 20 km (a) and 25km (b). For the purpose of comparison, the cumulative CARMA data between 0.02 - 0.1 μm are also shown.

Regarding OPC data, in most cases, a bimodal lognormal size distribution is used to fit OPC data if there are enough sizes measured (Deshler et al., 2003). However, the same approach may be not suitable for remote sensing instruments working in the limb geometry (Malinina et al., 2017). It is know that almost all OPC measurements have limitations below 0.1 μm such that the aerosol size distribution from 0.01 - 0.1 μm is poorly measured. The lack of information in the OPC data gap region would result in large uncertainty in calculating phase function. We have added an appendix, and Figs. A1-A2 in it speak to the point of this comment. In fact, the Wyoming in situ data just

updated and the minimum size measured is now claimed to be 0.094 µm (Deshler, private communication, 2018). We plan to use the new OPC data in future work.

*Page 7 L10: I am not sure what this means? "We find that the key difference between the two ASDs is that the Pueschel distribution has larger dN/dlogr values at 0.1 micron, which causes the derived aerosol scattering phase function P(Θ), shown in Figure 4, to be more "Rayleigh-like" at large single scattering angle , i.e., closer to the Rayleigh P(Θ)*

**Reply:** Reviewer 2 also noted this. More 'Rayleigh-like' means that comparing with CARMA phase function, Pueschel phase function is closer to Rayleigh phase function at larger Θ, i.e., closer to the Rayleigh P(Θ).

*Page 10: It may help the reader if authors could explain as how the extinction is computed and at what wavelengths the extinctions are calculated.*

**Reply:** Reviewer 2 made the same suggestion. The following explanatory text was now added: 'We retrieve aerosol extinction profiles at 675 nm from OMPS/LP radiance measurements. We first calculate the cross-section and phase function from the resulting Gamma aerosol size distribution using Mie theory, then run the radiative transfer model (RTM) within the aerosol retrieval code with the new cross-section and phase function to determine how the OMPS/LP aerosol retrieval changes with this new ASD relative to Pueschel ASD in the V1 ( Loughman et al., 2018).'

*Page 11 Figure 9: How does it look like in the lower stratosphere. This is where the main issue of all limb scatter measurements lies. I would like to see a similar plot for lower altitudes say 18, 16, or 13 km. If you could use a tropopause height climatology and use the above altitudes to do a similar plot, I expect to see some data showing up at 13 km for higher latitudes where I think limb scatter measurements have issues. What wavelength is extinction in Figure 9 calculated at?*

**Reply:** Agreed. We made the following figure showing that the ratio of extinctions (black dots) has a very large variability at 15.5km.

[Figure]

*Page 13 Figure 11: The figure says ASI's are computed at three different wavelengths. Are these wavelengths just used for ASI's or are these used for computing extinction as well (for example in Figure 9)?*

**Reply:** Figure 11 shows that ASI's are computed at six (we now changed 'three' to 'six') different wavelengths. These wavelengths just used for ASI's, not for computing extinction. All extinctions are computed at 675nm.

*Page 15 L15-20: The OSIRIS data are in reasonably good agreement with SAGEII (Rieger et al., 2015) except in the lower stratosphere at higher latitudes. I would like to see how CARMA ASD's derived extinction compare to OSIRIS. I understand it may be out of the scope of this paper but it would definitely help the stratospheric aerosol community as I believe OMPS measurements are valuable which may help fill the gap between SAGEII and SAGEIII-ISS in addition to OSIRIS/SCIAMACHY/CALIPSO.*

**Reply:** We have compared OMPS/LP and SAGEIII/ISS retrieved extinction profiles. The CARMA ASD's derived extinctions are in good agreement with SAGE III/ISS data. We have also compared OMPS/LP data to OSIRIS. We plan to publish the resulting analysis in the next paper.